# Characterization of Philadelphia-like Pre-B Acute Lymphoblastic Leukemia: Experiences in Mexican Pediatric Patients

**DOI:** 10.3390/ijms23179587

**Published:** 2022-08-24

**Authors:** Daniel Martínez-Anaya, Dafné Moreno-Lorenzana, Adriana Reyes-León, Ulises Juárez-Figueroa, Michael Dean, María Montserrat Aguilar-Hernández, Netzi Rivera-Sánchez, Jessica García-Islas, Victoria Vieyra-Fuentes, Marta Zapata-Tarrés, Luis Juárez-Villegas, Rogelio Paredes-Aguilera, Lourdes Vega-Vega, Roberto Rivera-Luna, María del Rocío Juárez-Velázquez, Patricia Pérez-Vera

**Affiliations:** 1Laboratorio de Genética y Cáncer, Instituto Nacional de Pediatría, Mexico City 04530, Mexico; 2Cátedra CONACYT-Instituto Nacional de Pediatría, Mexico City 04530, Mexico; 3Laboratorio de Citogenética, Instituto Nacional de Pediatría, Mexico City 04530, Mexico; 4Laboratory of Translational Genomics, Division of Cancer Epidemiology and Genetics, National Cancer Institute, Rockville, MD 20850, USA; 5Laboratorio de Genética Molecular, Hospital Infantil Teletón de Oncología, Queretaro 76140, Mexico; 6Coordinación de Investigación, Fundación IMSS, A.C., Mexico City 06600, Mexico; 7Servicio de Hemato-Oncología, Hospital Infantil de México Federico Gómez, Mexico City 06720, Mexico; 8Servicio de Hematología, Instituto Nacional de Pediatría, Mexico City 04530, Mexico; 9Dirección General, Hospital Infantil Teletón de Oncología, Queretaro 76140, Mexico; 10Servicio de Oncología, Instituto Nacional de Pediatría, Mexico City 04530, Mexico

**Keywords:** pre-B ALL, children, Ph-like, Mexican, *CRLF2* overexpression, pCrkl, *IGH::CRLF2*, *P2RY8::CRLF2*, iAMP21

## Abstract

Ph-like subtypes with *CRLF2* abnormalities are frequent among Hispano–Latino children with pre-B ALL. Therefore, there is solid ground to suggest that this subtype is frequent in Mexican patients. The genomic complexity of Ph-like subtype constitutes a challenge for diagnosis, as it requires diverse genomic methodologies that are not widely available in diagnostic centers in Mexico. Here, we propose a diagnostic strategy for Ph-like ALL in accordance with our local capacity. Pre-B ALL patients without recurrent gene fusions (104) were classified using a gene-expression profile based on Ph-like signature genes analyzed by qRT-PCR. The expressions of the *CRLF2* transcript and protein were determined by qRT-PCR and flow cytometry. The *P2RY8::CRLF2*, *IGH::CRLF2, ABL1/2* rearrangements, and Ik6 isoform were screened using RT-PCR and FISH. Surrogate markers of Jak2-Stat5/Abl/Ras pathways were analyzed by phosphoflow. Mutations in relevant kinases/transcription factors genes in Ph-like were assessed by target-specific NGS. A total of 40 patients (38.5%) were classified as Ph-like; of these, 36 had abnormalities associated with Jak2-Stat5 and 4 had Abl. The rearrangements *IGH::CRLF2,*
*P2RY8::CRLF2*, and iAMP21 were particularly frequent. We propose a strategy for the detection of Ph-like patients, by analyzing the overexpression/genetic lesions of *CRLF2*, the Abl phosphorylation of surrogate markers confirmed by gene rearrangements, and Sanger sequencing.

## 1. Introduction

The Philadelphia-like (Ph-like) subtype occurs in 10–15% of children diagnosed with precursor-B cell acute lymphoblastic leukemia (pre-B ALL) [1]. The World Health Organization has proposed the Ph-like subtype as a provisional entity, which requires additional efforts to clarify its significance and be incorporated into the accepted pre-B ALL classification [2]. Patients of this subset present multiple rearrangements, mutations, and copy number variations involving kinase or cytokine receptor genes, as well as activation of Jak2-Stat5, Abl, and Ras signaling pathways [1].

The Ph-like subtype is associated with a poor outcome, compared to other pre-B ALL subtypes (excluding *BCR-ABL1* and *KMT2A* rearrangements). Patients with Ph-like ALL show higher risks of induction failure, with high post-induction minimal residual disease levels when treated with conventional chemotherapy regimens. This poor early response translates into a lower overall survival [3,4,5]. However, preclinical studies and case reports indicate that Ph-like patients might benefit from clinical trials, including kinase inhibitors [1]. Despite the diversity of abnormalities in kinase genes identified in Ph-like patients, the majority (fortunately) converge in a limited number of signaling pathways that could be treated effectively using a combination of standard chemotherapy and Abl or Jak2/Stat5 class inhibitors [6].

The Ph-like subtype is particularly frequent (35%) among the Hispano–Latino high-risk pre-B ALL children residing in the USA. This prevalence has been associated with the Amerindian ancestry of this population [7]. In Mexico, ALL in patients <18 years old present with a high incidence, about 79.8 cases per million/year [8], and most have dismal prognoses since a large proportion (48.7% to 83.35%) of high-risk patients is referred [9]. The evidence of the increasing number of cases and the poor response to treatment suggests that the Ph-like subtype might be frequent in Mexican patients with pre-B ALL. Nevertheless, the genomic complexity of this subtype constitutes a challenge for accurate and fast diagnoses as it requires the use of diverse genomic methodologies [10,11].

To address the points outlined above, it is important to implement a diagnostic strategy for the identification of Ph-like genomic characteristics in Mexican patients with pre-B ALL. Nowadays, genomic technology, such as microarrays and next-generation sequencing, is not generally available at diagnostic laboratories in Mexico. Therefore, a new approach is needed to implement a diagnostic strategy, to identify Ph-like genomic characteristics in Mexican patients with pre-B ALL. Additionally, it is crucial that this diagnostic strategy relies on accessible methods for Mexican Health Institutions that treat patients with leukemia.

The present study aimed to develop a strategy, in accordance with our local capacity to diagnose Ph-like ALL children in Mexico, and describe the results obtained by using a number of methods to determine: (a) the expression of the *CRLF2* transcript and protein by qRT-PCR and flow cytometry; (b) the presence of *P2RY8::CRLF2*, *IGH::CRLF2, ABL1/2* rearrangements and the Ik6 isoform using RT-PCR and fluorescence in situ hybridization (FISH); (c) the positivity of a gene expression profile (GEP) determined by qRT-PCR; (d) the occurrence of surrogate markers of Jak2-Stat5/Abl/Ras pathways by phosphoflow; (e) the presence of mutations in relevant kinases/transcription factor genes in Ph-like by target-specific next generation sequencing (NGS) and Sanger sequencing.

## 2. Results

### 2.1. CRLF2 Expression and Genetic Lesions

The *CRLF2* overexpression was detected in 57 out of 104 patients. Across the 104 analyzed samples, the *CRLF2* genetic lesions detected were grouped in rearrangements, copy number variations, and point mutations (Figure 1, Appendix A).

(a)Forty-five patients with *CRLF2* rearrangements, thirty-five of them showed the *P2RY8::CRLF2* deletion, and ten cases with *IGH::CRLF2* translocation. Three of them showed two different cell clones, one with extra copies of the *IGH* and *CRLF2* loci as a result of hyperdiploidy, and one presented the coexistence of hyperdiploidy and *IGH::CRLF2* translocation (Figure 2A). Unfortunately, twenty-one patients with high expressions of *CRLF2*, including hyperdiploid cases, were not analyzed for *IGH::**CRLF2* since cell samples were not available.(b)Nine patients presented increased copy numbers of *CRLF2* as a result of hyperdiploidy involving sex chromosomes, and the array-CGH analysis showed one case with a *CRLF2* triplication without trisomy of sexual chromosomes (Figure 2B).(c)The *CRLF2* c.695T > G p.F23 activating point mutation was detected in one patient (Figure 2C).

**Figure 2 ijms-23-09587-f002:**
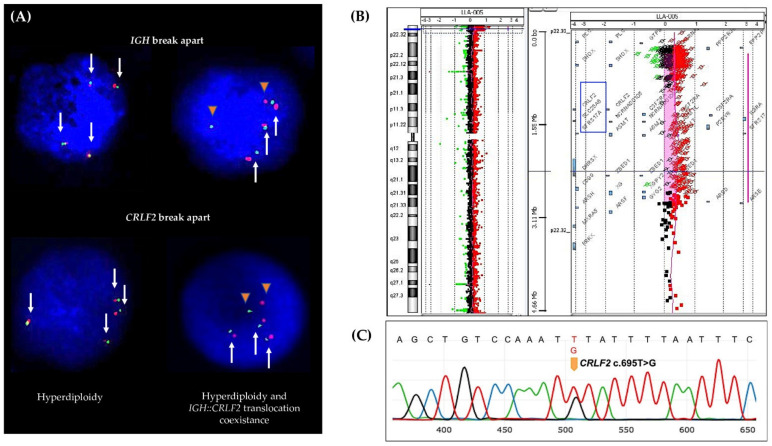
***CRLF2* genetic lesions.** (**A**) FISH assay showing cell clones with hyperdiploidy (left) and cell clones with *IGH::CRLF2* translocation and hyperdiploidy coexistence (right). White arrows represent the *IGH* and *CRLF2* loci and orange triangles represent the *IGH::CRLF2* translocation (fused signals green/red represent *IGH* or *CRLF2* normal loci, and separate green and red signals represent broken *IGH* or *CRLF2* loci). (**B**) aCGH shows a copy number gain at Xp22.33 (ChrX: 403328-2865335) involving the *CRLF2* gene. (**C**) Sanger sequencing shows the *CRLF2* point mutation.

### 2.2. Gene Expression Profile (GEP)

The GEP analysis was feasible in 101 patients. Among the patients with *CRLF2* overexpression, the GEP positive coefficients were presented as follows: (a) 0–0.57 GEP in 25 cases; (b) 0.67 in 10 cases, and (c) 0.71–1.0 in 19 cases. In contrast, patients with low/null *CRLF2* expression showed: (a) 0–5.7 GEP in 41 cases; (b) 0.67 in 6 cases, and (c) no patients with GEP of 0.71–1.0 (Figure 3).

A positive GEP (≥0.5) was detected in four patients with *CRLF2* overexpression (ALL 001, ALL 003, ALL 005, and ALL 008), which matched with the GEP established by Reshmi et al. 2017 as positive for the Ph-like subtype [12].

### 2.3. IKZF1 Abnormalities

The *IKZF1* dominant negative isoform Ik6 was analyzed in 100 patients and 55 were positive, one of them presented two mutations by NGS; 29/55 presented *CRLF2* overexpression (Figure 1). In three additional cases, deletion of the *IKZF1* gene was detected by array-CGH. This means that a total of 32 patients with *CRLF2* overexpression also presented *IKZF1* abnormalities. Regarding those patients with Abl pathway activation, in 7/10, *IKZF1* was abnormal (Figure 1).

### 2.4. Pathogenic and Likely Pathogenic Mutations in Kinase and Transcription Factor Genes

NGS and Sanger assays were performed on 47 patients. A total of 53 variants were detected. These variants are classified and described in Appendix A; of these, 32 pathogenic or likely pathogenic variants were observed (Figure 1). In descending order of mutation frequency per patient, the affected genes were: *NRAS* (8), *KRAS* (6), *PAX5* (4), *JAK2* (3), *NF1* (3), *IKZF1* (1), *PTPN11* (2), *SH2B3* (1), *JAK1* (1), *BRAF* (1), *FLT3* (1), *CSF1R* (1), and *USP9X* (1). A specific description is presented in Figure 1. No pathogenic and likely pathogen mutations were observed in *JAK3*, *EPOR*, *IL7R*, *IL2RB*, *ABL1*, *ABL2*, *PDGFRB*, *ETV6*, *NTRK3,* and *SSBP2* (Appendix A).

In 18 patients, only one mutation was detected (Figure 1). Four patients presented two mutations: (1) ALL 001 had two different mutations in *NRAS*, (2) ALL 010 had mutations in *JAK2* and *KRAS*, (3) ALL 017 had two different mutations in *SH2B3*, and (4) ALL 036 showed mutations in *NRAS* and *USP9X*. Two patients showed three mutations: (1) ALL 045 had one in *NRAS* and two different mutations in *IKZF1*, (2) ALL 046 showed two different mutations in *KRAS* and one in *CSFIR*. Finally, one patient presented four mutations: (1) ALL 002: one mutation in *JAK2* and *KRAS*, and two different mutations in *PAX5*.

Sanger sequencing revealed the *CRLF2* mutation c.695T > G (p.Phe232Cys) in patient ALL 001. This particular lesion is associated with the Ph-like subset. However, according to the ACMG criteria, it is considered a variant of ‘uncertain significance’ (Figure 1 and Appendix A).

### 2.5. CRLF2 Protein Surface, Phosphorylated Surrogate Markers, and Inhibition Analysis

Adequate samples for CRLF2 surface protein analysis were obtained in 39 patients. Four of them were positive and showed overexpression of the transcript as well as *CRLF2* rearrangements (three with an *IGH::CRLF2* translocation and one with a *CRLF2* triplication) (Figure 1).

The Jak2-Stat5 pathway was analyzed in 21 patients. Five of them showed activation (Figure 1 and Figure 4A). The overexpression and rearrangements of *CRLF2* were detected in four patients. Moreover, in two of these cases (ALL026 and ALL036), the phosphorylation of Stat5 was reverted using ruxolitinib.

In 17 patients, the Ras pathway was evaluated, showing a positive result in two patients (ALL 036 and ALL 040). Both cases were also positive for *NRAS* mutations, had high *CRFL2* gene expression, GEP ≥ 0.67, and *IKZF1* lesions. One of these patients was also positive to pCrkl (ALL 040), and case ALL 036 showed *IGH::CRLF2* rearrangement and phosphorylation of all three surrogate markers (Figure 1 and Figure 4A, Appendix A).

### 2.6. Patients with Features Associated with Ph-like and with B-Other Subtypes

Of the 57 patients with overexpression of *CRLF2*, 39 cases were considered to belong to the Ph-like subgroup (Figure 1):(a)All patients with *IGH::CRLF2*, cases with *P2RY8::CRLF2* deletion, and the single patient with triplication of CRLF2 [13].(b)All patients with mutations in *CRLF2*, *JAK1*, *JAK2*, and *SH2B3* [10,13].(c)All patients with GEP coefficients of 7.1–1.0, and the four cases who matched with the Reshmi et al. 2017 positive profile [12] were included in this subset.(d)All patients who were positive for the surrogate markers pStat5, pCrkl, and pErk [10]. Two more patients (ALL 060 and ALL 061) presented phosphorylation of surrogate markers; thus, they were also considered part of the Ph-like subgroup, although did not present *CRLF2* overexpression

The remaining 21 patients without adequate samples for the *IGH::CRLF2* analysis, underwent NGS or phosphoflow analysis. For these, mutations or activated pathways were detected in eight patients; however, six patients with hyperdiploidy and seven without abnormalities related to the subtype, were inconclusive. Thus, they were eligible for rearrangement and *JAK* mutation analyses since they displayed *CRLF2* overexpression (Figure 1). As for other abnormalities detected, three patients presented clonal *IGH* breakages and one was positive for Reshmi’s profile; unfortunately, it was not possible to determine the juxtaposed gene partner.

Other abnormalities detected in this study were found in 11 patients. These patients harbored the intrachromosomal amplification of chromosome 21 (iAMP21) and *P2RY8::CRLF2* deletion, six of them were part of the Ph-like subgroup.

The dominant negative isoform Ik6 was widely represented in all patients analyzed and was not particularly observed in patients with *CRLF2* overexpression.

Regarding NGS results, most of the RAS family mutations (75%), as well as the mutations detected in *SH2B3* and *CSF1R,* were presented in this group. We also observed the *PAX5* c.239C > G p.P80R mutation in two patients. Three patients with RAS mutations presented overexpression of *CRLF2* (ALL 020, ALL 027, and ALL 042, Figure 1, Appendix A); however, it was not possible to obtain a complete characterization to be considered Ph-like cases.

Among the 47 patients with lower or no expressions of *CRLF2* (Figure 1), 13 presented the *P2RY8::CRLF2* deletion, and five harbored iAMP21. In two of these five cases, the deletion was analyzed by FISH and showed a very low frequency (<0.5%). The GEP in this subset of patients was observed from 0.16 to 0.67, and in 15 patients it was zero.

In summary, in our population, 40 of 104 patients were classified as Ph-like, 36 presented abnormalities associated with Jak2-Stat5, and four had Abl. In this experience, the following features were decisive to detect Ph-like patients (Figure 5):(a)For the Jak2-Stat5 class, the presence of *CRLF2* overexpression and rearrangements, JAK family mutations, and the Jak2-Stat5 activation pathway.(b)For ABL class, the presence of the pCrkl surrogate marker and the following detection of *ABL1* or *ABL2* rearrangements.(c)Despite few patients being analyzed with the pErk, those cases tested for the Ras pathway and mutations showed consistent results.

**Figure 5 ijms-23-09587-f005:**
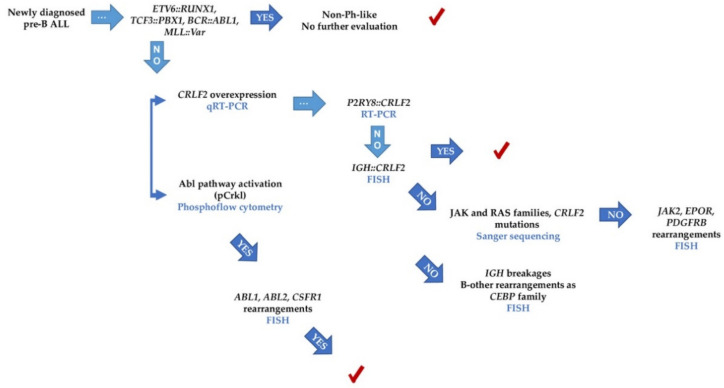
**Adapted strategy for Ph-like diagnosis.** Proposed testing strategy for the Ph-like subgroup. The analysis comprises the pre-B ALL patients without recurrent gene fusions. ✔ = no further evaluation since a final result was obtained.

## 3. Discussion

Ph-like or *BCR-ABL1*-like ALL was discovered in two independent and partially overlapping GEPs and cohorts [14,15,16]; however, although some patients can be classified as discordant, they are currently being considered as a single subtype. The main sources of disparity include: methodological differences in gene expression analysis, different compositions of pre-B ALL subtypes, risk groups, and the genetic ancestries of the patients [17]. Based on this, we can infer that the different GEPs might be determined by the genetic backgrounds of the studied population. More recently, diverse approaches for detecting Ph-like ALL have been described, including the reduction of the genes that integrate GEPs combined with the screening of *CRLF2/JAK* and Abl type gene rearrangements, or the determination of the *CRLF2* overexpression followed by the direct detection of the most frequent genomic abnormalities [12,18,19,20]. These improvements aim to simplify the routine diagnosis of Ph-like ALL patients; however, the plethora of genomic abnormalities in this subtype requires extensive efforts as well as the application of diverse genomic methods to become feasible. Furthermore, even after combining diverse genomic methods, a group of patients might show a lack of kinase-activating abnormalities [11]. Recently, different strategies based on the RNA-sequencing technology have been proposed as useful tools to detect known and novel gene fusions, several clinically relevant genetic alterations, sequence mutations, and gene expression profiles in a single assay. For these reasons, it has been proposed to integrate this technology into front-line assays for childhood ALL [21]. Unfortunately, these methods are not always available in the laboratories of Institutions that treat childhood cancer in Mexico.

As for Hispano–Latino patients residing in the USA, Harvey et al., 2010, demonstrated that *CRLF2* abnormalities are particularly frequent in high-risk ALL children [7]. Raca et al. 2017 determined that *IGH::CRLF2* is the most common rearrangement in Hispano–Latino ALL children from Los Angeles [20]. Similar results were obtained by our group as we observed that *CRLF2* overexpression and rearrangements are frequent in Mexican pre-B ALL children [22]. In the present study, the strategy to detect Ph-like patients was suitable to identify *CRLF2* abnormalities; in most of the cases, it was possible to associate the *CRLF2* overexpression with a rearrangement, point mutation, and/or an extra copy of the gene (Figure 1). One limitation of our study is that it was not possible to classify a small subset of cases as Ph-like patients due to the presence of multiple copies of the *CRLF2* locus, secondary to hyperdiploidy, since it is controversial if these cases should be included in the subgroup [19]. Nevertheless, we could not discard them since they presented a *CRLF2* overexpression, which requires the *IGH::CRLF2* characterization.

In contrast with ALL Hispanic children from Los Angeles [20], in the present study, the *P2RY8::CRLF2* deletion was more frequent than the *IGH::CRLF2* rearrangement; whereas, our *IGH::CRLF2* frequency was close to the one reported by the mentioned group (9.6% vs. 12%, respectively), and it might be higher, considering that 21 patients with *CRLF2* overexpression could not be included due to the lack of cell samples available for an additional FISH analysis. Herein, we found that *P2RY8::CRLF2* was present in 33% of patients; this high occurrence likely represents a primary lesion, which could be considered a Ph-like subtype-defining feature in patients with high *CRLF2* overexpression, but sometimes this may also represent a secondary aberration or a co-occurrence with an established primary lesion, such as the iAMP21 [23]. Notably, in this study, we observed a high incidence (10.5%) of iAMP21, compared with reports from other ALL populations (1–2%) [24]; this disparity could influence the increased frequency of *P2RY8::CRLF2* as a secondary abnormality, especially when it is detected in a low number of cells and does not result in a *CRLF2* overexpression [24,25,26]. In the context of secondary abnormalities, the mutation *PAX5* c.239C > G p.P80R was observed in two patients. This abnormality has been considered a subtype defining alteration associated with a favorable prognosis. It has been reported in cases that lacked any fusion genes [27]. Nevertheless, in our patients, this mutation may be considered a secondary oncogenic event, because one patient harbored *IGH::CRLF2,* an *IKZF1* biallelic alteration, and *NRAS* mutation, and the second case showed *P2RY8::CRLF2* and a *PTPN11* mutation.

Early diagnosis of Ph-like ALL is crucial to defining risk stratification and optimizing therapeutic strategies focused on the incorporation of targeted therapies into chemotherapy schedules [28]. The abnormalities of the Jak2-Stat5 pathway are frequent in Ph-like ALL; however, although most of the Ph-like patients (80%) do not present Abl type fusions, it is important to detect Abl positive cells since they are sensitive to currently available tyrosine kinase inhibitors (TKIs) [10]. As an example, Tanasi et al., 2019, reported an experience in a small cohort of Ph-like ALL patients positive to Abl-class fusions in which the introduction of TKIs during consolidation improved the post-induction minimal residual disease (PI-MRD) negative status with values below 10^−4^, compared with the PI-MRD ≥ 10^−2^ value observed in patients without TKIs-added chemotherapy. This was associated with a 3-year overall survival improvement of 77% [29].

There are challenges in the diagnosis of Ph-like ALL before using a targeted therapy treatment. Considering that NGS is not widely available in our diagnostic laboratories, we relied on the analysis of surrogate markers of signaling pathways, such as pCrkl [10]. The activations of Abl and Jak2-Stat5 pathways were detected in a total of 9.6% of patients, and 6.7% presented evidence of *CRLF2* abnormalities. This was an expected finding since it has been reported that Ph-like patients of the ABL class can show constitutive pCrkl and pStat5 [30]. Another possible explanation could be the coexistence of an Abl type gene fusion with *CRLF2* rearrangements, different from *BCR-ABL1*, the latter was observed in patient ALL 046, who was positive for *IGH::CRLF2* and *ABL2* breakage. The coexistence of *BCR-ABL1* with *IGH::CRLF2* or *P2RY8::CRLF2* has been reported in a subset of patients diagnosed as Ph-positive; this feature has been related to relapses and treatment resistance. Although these findings are uncommon, they seem to be more frequent in patients of Hispanic and Mexican–American origins [31,32]. Similar to previous reports, RAS family mutations and pathway activations were frequently detected in patients with Jak2-Stat5/*CRLF2* or Abl pathway abnormalities [30].

Based on our findings, the patients that showed overexpression of the *CRLF2* transcript in addition to the *CRLF2* rearrangement and a GEP ≥ 0.71 fit with the Ph-like subtype. However, it is important to note that the GEP only provided certain guidance on the possible association of GEP in the Ph-like subset as false positives and negatives were detected. The selected genes of our GEP were based on those frequently reported in Ph-like signatures since the eight-gene standard GEP [12] had not yet been reported at the time of this study; nevertheless, even this signature has its own limitations [11]. According to our results, we propose analyzing the patients without recurrent pre-B ALL gene fusions, searching for *CRLF2* overexpression by qRT-PCR. We suggest that patients with this characteristic should be screened for *P2RY8::CRLF2* and, in case of a negative result for this deletion, the *IGH::CRLF2* rearrangement must be investigated by FISH. Furthermore, the screening for Abl activation must be performed in all patients, especially in those without evidence of *CRLF2* rearrangements (Figure 5).

In the present study, it was possible to use genomic methodology, such as array-CGH and targeted specific NGS, which allowed us to know part of the genomic landscape of our Ph-like patients. However, the JAK family-activating lesions, such as *JAK2* or *EPOR* gene rearrangements, or the ABL class lesions, such as *PDGFRA/B* or *CSF1R* gene fusions, are not detectable using those genomic approaches. Remarkable, since RNA-sequencing is an unavailable technology in our country, different strategies based on FISH, multiplex RT-PCR, or flow cytometry protocols could be used in order to identify targetable gene fusions and abnormal signaling pathway activation in Ph-like patients [33,34].

However, one concern arises from this experience as we need to adapt the Ph-like screening to a feasible strategy for the laboratories dedicated to diagnosing ALL patients in Mexico. We believe that our strategy, concerning PCR, flow cytometry, FISH, and Sanger sequencing, is fundamental and possible to perform in most of the laboratories that diagnose and classify ALL. However, this strategy has its own limitations and could be improved to obtain a better ALL classification. In order to implement a better strategy, and based on our results and the literature recommendations, we propose that the following methods be implemented: (a) Sanger sequencing focused on the recurrent mutated exons for genes concerning *JAK* and *RAS* families, and for *CRLF2*; (b) searching for rearrangements in *JAK2, EPOR, PDGFRB*, and possible partners of the *IGH* rearrangements associated with B-other ALL (CEBP gene family); (c) in patients with Abl activation: to include at least one more surrogate marker (pSrc or pStat5), and searching for rearrangements in *ABL1, ABL2*, and *CSFR1*; (d) assays with and without inhibition must be performed to complement the Jak2-Stat5 and Abl pathway analyses, to address the differences found in activation pathways and in the response to kinase inhibitors, which have been observed between diverse gene fusions [30].

Despite this study not being designed to obtain the frequency of the Ph-like subtype in our population, due to the high incidence of *CRLF2* abnormalities found, we can suggest that this subset is well represented in Mexican pre-B ALL patients. However, rearrangements of *CRLF2* can also be found in non-Ph-like patients [22] and, thus, this may affect the true frequency of Ph-like patients in this study.

Pre-B ALL patients should be studied at least with the most basic strategy proposed above. In addition to this, we observed a high frequency of iAMP21 in our patients, frequently associated with *P2RY8::CRLF2* deletion, as well as the presence of *IKZF1* abnormalities. Collectively, our results appear consistent with the ones previously reported in the literature [20,35,36]; hence, the detection of these lesions must be considered in the routine screening of genetic abnormalities in pre-B ALL in Mexican patients. Finally, as all these abnormalities require high-risk therapy regimens, which include the use of specific inhibitors, a mandatory routine identification of this genetically heterogeneous entity will have a positive contribution to the overall survival of Mexican patients [1,11].

## 4. Materials and Methods

### 4.1. Editorial Policies and Ethical Considerations

Patients or their guardians signed an informed assent/consent to participate, in compliance with the ethical principles enunciated by the Declaration of Helsinki. The participant Institutions’ Research and Ethics Committees approved this study (project numbers 067/2014 and 019/2016 with the National Commission of Bioethics registration number CONBIOETICA-09-CEI-025-20161215).

### 4.2. Patients and Samples

Bone marrow samples from 104 pre-B ALL patients ≤18 years old were collected at diagnosis (102) or relapse (two), from January 2015 to December 2018. As previous reports recommend, patients with the recurrent gene fusions *ETV6::RUNX1, TCF3::PBX1, KMT2A*-var, and *BCR::ABL1* were excluded, since these subtypes no longer coexist with the Ph-like subset [15,16,17]. Patients were recruited at the National Institute of Pediatrics (INP-Instituto Nacional de Pediatría), Federico Gómez Children’s Hospital of Mexico (HIMFG–Hospital Infantil de México Federico Gómez, Mexico City, Mexico) both in Mexico City, and the Children’s Hospital Teleton of Oncology (HITO- Hospital Infantil Teletón de Oncología, Querétaro, Mexico) in Querétaro State.

Patients were classified using a gene expression profile (GEP) constituted by the reported genes of the Ph-like signature [14,15,16,17,18,37]. The expressions of the *CRLF2* transcript and protein were determined. The most frequent *CRLF2* genetic rearrangements, *P2RY8::CRLF2* and *IGH::CRLF2*, as well as the characteristic *CRLF2* point mutation c.695T > G (p.Phe232Cys), were evaluated. The *IKZF1* dominant negative isoform Ik6 was also detected as one of the hallmarks of the Ph-like subset [15], and patients with available samples were screened for the pathogenic variants of the most relevant kinases and transcription factors in this subtype: *JAK1, JAK2, JAK3, EPOR, IL7R, IL2RB, KRAS, NRAS, PTPN11, BRAF, NF1, FLT3, ABL1, ABL2, PDGFRB, CSF1R, ETV6, NTRK3, IKZF1, PAX5, SH2B3, SSBP2*, and *USP9X* [30,38]. In six cases, array-CGH was performed to obtain complementary information. When samples were found adequate, these studies were integrated with a flow cytometric phosphorylation analysis of the Jak2-Stat5, Abl, and Ras pathway targets. To confirm the obtained results, the phosphorylation of targets was reverted using specific pathway inhibitors.

### 4.3. Analysis of GEP

We analyzed the gene expression profiles constituted by six or seven genes (*CRLF2*, *TSPN7*, *IGJ*, *PON2*, *SEMA6A*, *BMPR1B*, and/or *MUC4*) frequently included in previously reported Ph-like signatures [14,15,16,17,18,37]. The expressions of the first six genes were determined in 54 patients; in 47 cases, *MUC4* was also analyzed; in three patients, the samples were not adequate for testing; *GUSß* was used as the endogenous control. In brief, isolated mononuclear cells from bone marrow samples were processed for RNA extraction (RNeasy kit. Qiagen, Düsseldorf, Germany) and cDNA was obtained with standard methods (Invitrogen, Waltham, MA, USA). The relative expression of each gene of the profile was determined in duplicate by qRT-PCR (LightCycler 2.0 Instrument; Roche Applied Science, Penzberg, Upper Bavaria, Germany) using TaqMan gene expression probes from the Universal Probe Library System (Roche Applied Science, Penzberg, Upper Bavaria, Germany). The primer sets and probes for the genes of this assay are presented in Appendix A. The cut-off value for the expression of each transcript was based on the analysis of quartiles and was set as previously described [36,39].

### 4.4. Determination of the CRLF2 Protein

The protein expression was determined by flow cytometry in 39 suitable samples. Bone marrow cells were incubated with monoclonal antibodies against CD45 Amcyan, CD34 PECy7, CD3 PerCP, MPO PE, CD13 or CD7 FITC, and CRLF2 APC (BD, Franklin Lakes, NJ, USA). The cell surface CRLF2 protein was assessed on both fresh and fixed cells (BD Cytofix/Cytoperm, Franklin Lakes, NJ, USA) using a BD FACSVersa Cell Analyzer System (BD, Franklin Lakes, NJ, USA), and the data analysis with FlowJo v10 software for Mac OS X (BD, Franklin Lakes, NJ, USA).

### 4.5. Detection of P2RY8::CRLF2 Deletion

*P2RY8::CRLF2* was determined by RT-PCR in all patients based on the methods described by Palmi et al., 2012 [25]. A second amplification round was performed using the same set of primers and the PCR amplicon of the first reaction as the template. Patients without suitable samples to perform the RT-PCR technique were studied by FISH; at least 200 interphase nuclei and metaphases were analyzed with a PAR1 deletion probe (CytoCell–OGT, Oxford, UK) following the manufacturer’s recommendations.

### 4.6. Detection of the IGH::CRLF2 Rearrangement and ABL1 and ABL2 Breakages

*IGH::CRLF2* was determined in patients with *CRLF2* overexpression and negative to *P2RY8::CRLF2* deletion. This rearrangement was analyzed by FISH on 200 interphase nuclei and metaphases, using the dual-color break-apart probe LSI *IGH* (Abbott Molecular, Chicago, IL, USA) and the dual-color break-apart probe *CRLF2* (CytoCell–OGT, Oxford, UK), following the manufacturer’s recommendations. The patients with *IGH::CRLF2* rearrangement presented an equivalent proportion of cells with broken signals with each probe.

*ABL1* and *ABL2* breakages were determined in patients positive for the pCrkl marker. For *ABL1,* the *BCR-ABL1* dual color dual fusion FISH probe (QBiogene, Montreal, QC, Canada) and the *ABL2* break-apart probe (Cytotest ^TM^, Rockville, MD, USA), were used following the manufacturer’s recommendations. At least 200 cells were analyzed in each assay.

### 4.7. Analysis of Ik6 Transcript

The deletion of coding exons 3 through 6 result in the expression of the dominant-negative isoform Ik6 of *IKZF1* [15]. The transcript of Ik6 was determined in all patients with an available sample and detected by nested RT-PCR based on the methods described by Lin et al., 2016 [40]. The Ik6 expected size of the PCR product was 255 bp.

### 4.8. Target Specific NGS and Sanger Sequencing

NGS was performed for the analysis of the exome of the previously referred genes. The isolated mononuclear cells from bone marrow samples were processed for DNA extraction (DNeasy ^®^ Blood & Tissue kit. Qiagen, Düsseldorf, Germany). The DNA was sequenced on an Illumina HiSeq 2000 platform according to the manufacturer’s instructions, and two paired reads were generated for each sample. An in-house script for QC, align, calibration, and annotation of variants was used for the sequencing reads, followed by the analysis with the Genome Analysis Toolkit [41]. The NCBI human reference genome (hg19) was used. Next, a filter variant process was included for searching missense variants, in-frame insertions and deletions, frameshift, stop gain, and loss variants.

The exclusion of polymorphic variants was based on the population allele frequency values reported in GnomAD Exomes American Admixed/Latinos, GnomAD Exomes Global, or dbSNP (1000G or ExAC) databases. The interpretation of sequence variants was based on recommendations of the American College of Medical Genetics and Genomics, and the Association of Molecular Pathology [42]; the mutations considered in this study were those classified as pathogenic and likely pathogenic. Representative mutations detected by NGS and the *CRLF2* point mutation, c.695T > G (p.Phe232Cys), were determined by Sanger sequencing based on Yoda et al., 2010 [43] (Appendix A).

### 4.9. Array-CGH Analysis

DNA from bone marrow was used to process an Oligo 60 K array Agilent Technologies TM (Santa Clara, CA, USA), and was annotated using Human Genome Build GRCh37/hg19. It was performed according to standardized protocols provided by the manufacturer.

### 4.10. Kinase Activation and Inhibition Assay by Phosphoflow Cytometry

To detect Jak2-Stat5 pathway activation, fresh cells from bone marrow were stimulated with 200 ng/mL of thymic stromal lymphopoietin (TSLP) (PeproTech, Cranbury, NJ, USA) for 30 min at 37 °C, and stained with the following cell surface monoclonal antibodies: CD45 Amcyan, CD34 PECy7, CD19 PerCP, and CD10 PE (Becton Dickinson, Franklin Lakes, NJ, USA). Then, the cells were fixed and incubated with monoclonal antibodies, as previously reported, to identify the phosphorylated target Stat5 (pY694) Pacific Blue. For Abl and Ras pathways, similar conditions were used, except for the TSLP stimulation; the surrogate marker CrkL (pY207) AF488 was used for Abl, and Erk 1/2 (pT202/pY204) AF488 for Ras; a positive result was considered when populations >5% of blasts were detected. In samples with adequate numbers of cells, abnormal pathway activity was selectively inhibited in vitro for 30 min with ruxolitinib (5 nM) (Selleckchem, Houston, TX, USA) to inhibit the Jak2-Stat5 pathway, and with imatinib (5 μM) (Selleckchem) to prevent Abl abnormal activation. The antibody concentration and conditions for each assay were performed based on the manufacturer’s recommendations. Flow cytometry was performed using a BD FACSVerse Cell Analyzer System and data were analyzed by FlowJo vX software.

## Figures and Tables

**Figure 1 ijms-23-09587-f001:**
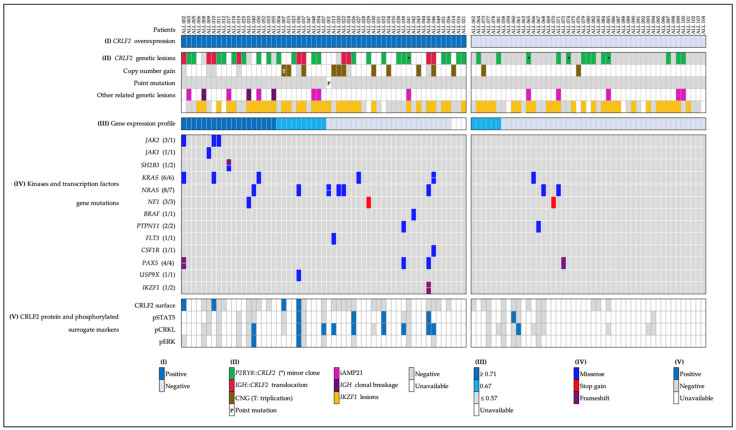
**Genetic landscape of pre-B ALL cases clustered according to the *CRLF2* expression.** Each column represents a patient, and the rows represent genetic or biochemical features. Gene mutations are shown in parenthesis as follows: patients harboring the mutation/different mutations identified. CNG: copy number gain.

**Figure 3 ijms-23-09587-f003:**
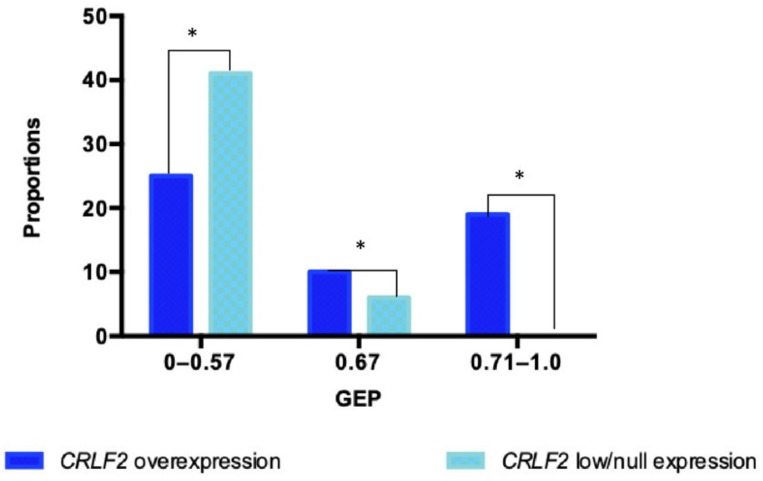
***CRLF2* expression distribution between the different GEP coefficients.** Different GEP coefficients were compared between patients with and low/null *CRLF2* overexpression, statistic differences are presented (* *p* < 0.05 chi-square test).

**Figure 4 ijms-23-09587-f004:**
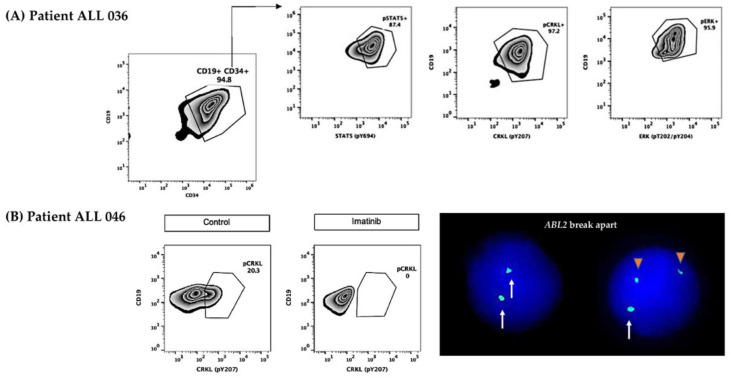
**Two patients with signaling pathway activation.** (**A**) Patient ALL 036 showing positivity to the three surrogate markers pStat5, pCrkl, and pErk. (**B**) Patient ALL 046 showing positivity to pCrkl and reversion produced by imatinib, and interphase nuclei hybridized with *ABL2* break-apart FISH probe with a normal pattern (two large green signals in the left nucleus) and with an *ABL2* breakage (one large green signal representing the normal gene, and two small green signals representing the broken gene in the rightright nucleus). White arrows represent the normal *ABL2* loci and orange triangles represent the breakage of *ABL2*.The Abl pathway was assessed in 29 patients and the surrogate marker pCrkl was positive in 10 (Figure 1 and Figure 4A); of these, 7 patients presented a genetic abnormality involving *CRLF2*, interestingly, 4 out of 7 cases were also positive for pStat5. On the other hand, three of the seven cases showed no evidence of abnormalities in *CRLF2* and the GEPs were 3.3, 6.7, and 1.0. It was possible to analyze two patients positive to pCrkl (ALL 039 and ALL 046) with *ABL1* and *ABL2* break-apart FISH probes, positive results were obtained for *ABL2* breakage in 16% and 6% of cells analyzed, respectively. Interestingly, patient ALL 046 was also found to be positive for *IGH::CRLF2* and showed the presence of both surrogate markers pStat5 and pCrkl, revealing the coexistence of both abnormalities in the same patient. Remarkably, this case showed pCrkl reversion using imatinib (Figure 1 and Figure 4B).

## Data Availability

Not applicable.

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
