# Peer review of "Characterization of Philadelphia-like Pre-B Acute Lymphoblastic Leukemia: Experiences in Mexican Pediatric Patients"

_ijms, 2022, doi:10.3390/ijms23179587_

Round 1
Reviewer 1 Report
In this study, Martínez-Anaya and co-authors used multiple methodologies to diagnose Ph-like ALL in accordance with their local capacity in Mexico. These approaches included expression of genes known to be part of the Ph-like signature as previously described by others, analysis of expression of the CRLF2 transcript and protein by qRT-PCR and flow cytometry, respectively. Moreover, common gene fusions were screened using RT-PCR and FISH. Surrogate markers of Jak2-38 Stat5/Abl/Ras pathways were analyzed by Phosphoflow. Mutations in relevant kinases/transcription factors genes in Ph-like were assessed by target-specific NGS.
Major comments:
Results:
- It would be helpful for the reader having a Supplementary table reporting in each raw an individual patient and in columns clinical characteristics and molecular results.
- Page 6: In addition to the Ras pathway activation/and or pCrkl, which alterations were present in patients ALL 036 and ALL 040?
- Page 7, lines 208-210: “We also observed the 208 PAX5 c.239C>G p.P80R mutation in two patients, but they were not included in the description because they are not considered as pathogenic or likely pathogenic”. This statement is not correct since PAX5 P80R is a pathogenic variant.
- Page 7, line 217: Ras alteration only is not a criteria for diagnosing Ph-like
- Genes throughout the text should go in italics
- The authors do not discuss here common fusions, such as JAK2, EPOR, PDGFRB and NTRK3 fusions.
Discussion:
One of the most comprehensive approach is represented by RNA-seq since it can detect known and unknown fusions activating kinase signaling pathways and can confirm clustering with Ph-positive samples. This approach should be at least discussed in the Discussion.
Methods:
- Page 10: “Patients were classified using a gene expression profile (GEP) constituted by the reported genes of the Ph-like signature”. Please add a reference.
Author Response
To the Reviewers,
Thank you for your comments and suggestions. Find below the answers to the questions. You can find all the corrections in the manuscript highlighted in yellow.
Reviewer one
Major comments:
Results:
- It would be helpful for the reader having a Supplementary table reporting in each raw an individual patient and in columns clinical characteristics and molecular results.
The original Table S2 was substituted by the new suggested table (Table S1) in supplementary material. The new Table S1 is cited in pages 3 and 7. The new Table S2 (previously Table S1) was reordered and cited in pages 5 and 6.
- Page 6: In addition to the Ras pathway activation/and or pCrkl, which alterations were present in patients ALL 036 and ALL 040?
The additional alterations present in patients ALL 036 and 040 patients were included in page 7.
- Page 7, lines 208-210: “We also observed the 208 PAX5 c.239C>G p.P80R mutation in two patients, but they were not included in the description because they are not considered as pathogenic or likely pathogenic”. This statement is not correct since PAX5 P80R is a pathogenic variant.
The classification of mutation 208 PAX5 c.239C>G p.P80R was corrected in page 7, Table S2 and Figure 1. The number of patients with pathogenic variants was corrected in the text (32) and considered in the description of these variants in all the text. (Pages 5 and 7)
- Page 7, line 217: Ras alteration only is not a criteria for diagnosing Ph-like
The RAS only definition was reconsidered and eliminated, based on this, the number and description of patients pertaining to the Ph-like subtype was modified. (Abstract, pages 7 and 8)
- Genes throughout the text should go in italics
This was reviewed and modified. (Page 12)
- The authors do not discuss here common fusions, such as JAK2, EPOR, PDGFRB and NTRK3 fusions.
A comment about this point was added in discussion in page 10.
Discussion:
One of the most comprehensive approach is represented by RNA-seq since it can detect known and unknown fusions activating kinase signaling pathways and can confirm clustering with Ph-positive samples. This approach should be at least discussed in the Discussion.
A paragraph about RNA-seq and its advantages was added in Discussion (Page 9: “Recently, different strategies based on the RNA-sequencing technology… “)
Methods:
- Page 10: “Patients were classified using a gene expression profile (GEP) constituted by the reported genes of the Ph-like signature”. Please add a reference.
The references were added. (Page 11)
Reviewer 2 Report
The present study was undertaken to develop a strategy for the differential diagnosis of Ph-like ALL in children in Mexico. It is important to note that in their experimental design, the authors sought to build on local Mexican capabilities, using a range of available methods to identify more characteristic genetic abnormalities in Mexican children. As a result of their research, the authors propose a strategy for detecting Ph-like patients by analyzing CRLF2 overexpression/genetic lesions, Abl-phosphorylation of surrogate markers confirmed by gene rearrangements, and Sanger sequencing. The study is well designed and presented.
At the same time, I would recommend expanding the Introduction and Discussion sections by providing more compelling rationale and evidence for how early, accurate differential diagnosis of Ph-like pre-B ALL can improve management and overall outcomes in pediatric patients.
Author Response
To the Reviewers,
Thank you for your comments and suggestions. Find below the answers to the questions. You can find all the corrections in the manuscript highlighted in yellow.
Reviewer two
I would recommend expanding the Introduction and Discussion sections by providing more compelling rationale and evidence for how early, accurate differential diagnosis of Ph-like pre-B ALL can improve management and overall outcomes in pediatric patients.
The introduction was improved based on your suggestions. (Page 2, 2nd paragraph)
The discussion was expanded based on your suggestions. (Page 9, 4th paragraph)
Round 2
Reviewer 1 Report
The authors addressed all my previous comments. I would have only one additional suggestion of mentioning in the Discussion that rearrangements of CRLF2 can be found also in non-Ph-like patients and thus this may affect the true frequency of Ph-like patients in this study.
